# Investigation of the Effects of Monomeric and Dimeric Stilbenoids on Bacteria-Induced Cytokines and LPS-Induced ROS Formation in Bone Marrow-Derived Dendritic Cells

**DOI:** 10.3390/ijms24032731

**Published:** 2023-02-01

**Authors:** Peter Riber Johnsen, Cecilia Pinna, Luce Mattio, Mathilde Bech Strube, Mattia Di Nunzio, Stefania Iametti, Sabrina Dallavalle, Andrea Pinto, Hanne Frøkiær

**Affiliations:** 1Department of Veterinary and Animal Sciences, University of Copenhagen, Ridebanevej 9, 1871 Frederiksberg, Denmark; 2Department of Food, Environmental and Nutritional Sciences (DeFENS), University of Milan, Via Celoria 2, 20133 Milan, Italy

**Keywords:** monomeric stilbenoids, dimeric stilbenoids, immune modulating, ROS formation, dendritic cells

## Abstract

Stilbenoids are anti-inflammatory and antioxidant compounds, with resveratrol being the most investigated molecule in this class. However, the actions of most other stilbenoids are much less studied. This study compares five monomeric (resveratrol, piceatannol, pterostilbene, pinostilbene, and trimethoxy-resveratrol) and two dimeric (dehydro-δ-viniferin and *trans*-δ-viniferin) stilbenoids for their capability to modulate the production of bacteria-induced cytokines (IL-12, IL-10, and TNF-α), as well as lipopolysaccharide (LPS)-induced reactive oxygen species (ROS), in murine bone marrow-derived dendritic cells. All monomeric species showed dose-dependent inhibition of *E. coli*-induced IL-12 and TNF-α, whereas only resveratrol and piceatannol inhibited IL-10 production. All monomers, except trimethoxy-resveratrol, inhibited *L. acidophilus*-induced IL-12, IL-10, and TNF-α production. The dimer dehydro-δ-viniferin remarkably enhanced *L. acidophilus*-induced IL-12 production. The contrasting effect of resveratrol and dehydro-δ-viniferin on IL-12 production was due, at least in part, to a divergent inactivation of the mitogen-activated protein kinases by the two stilbenoids. Despite having moderate to high total antioxidant activity, dehydro-δ-viniferin was a weak inhibitor of LPS-induced ROS formation. Conversely, resveratrol and piceatannol potently inhibited LPS-induced ROS formation. Methylated monomers showed a decreased antioxidant capacity compared to resveratrol, also depending on the methylation site. In summary, the immune-modulating effect of the stilbenoids depends on both specific structural features of tested compounds and the stimulating bacteria.

## 1. Introduction

Stilbenoids, which include resveratrol (3,4′,5-trihydroxy-*trans*-stilbene) and its hydroxylated (e.g., piceatannol) and methylated (e.g., pterostilbene and pinostilbene) derivatives, are a group of naturally occurring phenolic compounds found in various plant species. Stilbenoids possess several biological activities, including anti-inflammatory, antioxidative, and anticarcinogenic properties [1]. The bioavailability of the different stilbenoid compounds differs substantially and may depend on multiple factors, such as solubility, lipophilicity, and metabolic and chemical stability [2]. Substitution of hydroxy with methoxy groups increases lipophilicity, so that pterostilbene is more bioavailable than resveratrol [1]. Furthermore, resveratrol-derived dimers, such as δ-viniferin and its dehydro analogue, are also characterized by different bioavailability and biological activities compared to their monomers. For these reasons, information on the relative potency of stilbenoids, including whether a relationship exists between the anti-inflammatory and antioxidant ability, is still fuzzy.

Macrophages and dendritic cells constitute key cell types involved in inflammation and in the initiation of immune responses. In particular, dendritic cells play an essential role in the conduction of immune responses, e.g., by activating different subtypes of T-cells through the production of cytokines in response to signals provided by microbial organisms [3]. Regarding bacteria-induced cytokine production, it has been previously demonstrated that Gram-negative and Gram-positive bacteria induce responses through different pathways [4,5,6].

The antioxidant effect of stilbenoids has often been assumed to explain most of the other observed bioactivities, not least the anti-inflammatory effect [7]. However, stilbenoids have also demonstrated the ability to act on a broad range of cellular molecules involved in inflammatory signaling [8]. Most studies on the antioxidative or anti-inflammatory properties of stilbenoids were conducted on resveratrol, whereas only marginal attention has been placed on its hydroxylated and methylated analogues [8]. Additionally, assessment of bioactivities has generally been performed in a limited experimental setup without considering that the modulation exerted by these molecules is dependent on the nature of the stimulating agent [4,5].

To our knowledge, no systematic study comparing the immune-modulating and antioxidant effects of the various stilbenoid subclasses has been conducted so far. Here, a comparative and inclusive study has been performed to assess the immune-modulating effect of monomeric (resveratrol, piceatannol, pterostilbene, pinostilbene, and trimethoxy-resveratrol) and dimeric (dehydro-δ-viniferin and *trans*-δ-viniferin) stilbenoids [9] on the production of bacterially induced cytokines (IL-12, IL-10, and TNF-α) in murine bone marrow-derived dendritic cells (bmDCs). The study also included the possible effects on the lipopolysaccharide (LPS)-induced production of reactive oxygen species (ROS) in the same experimental system. In this frame, the relationship between the aforementioned effects and the antioxidant capacity of the individual molecules (as evaluated by ABTS and DPPH assays) was also investigated to provide a comprehensive vision of these systems.

## 2. Results

### 2.1. Methylation of Resveratrol Diminishes Its Inhibitory Effect on the L. Acidophilus NCFM-Induced Cytokine Response

Murine bmDCs stimulated with an increasing multiplicity of infection (MOI) of *L. acidophilus* NCFM responded with a dose-dependent production of IL-12, IL-10, and TNF-α (Figure 1A). An increasing concentration of either stilbenoid resulted in a dose-dependent inhibition of all three cytokines, leading to at least a 70% decrease in their production at the highest stilbenoid concentration.

Methylated analogues demonstrated a decreased inhibition capacity compared to resveratrol. More specifically, methylation of two hydroxy groups in resveratrol (i.e., pinostilbene and pterostilbene) resulted in a less decreased cytokine production, while methylation of all three hydroxy groups in resveratrol (i.e., trimethoxy-resveratrol) almost entirely abrogated the inhibitory capacity (Figure 1, panels D–F). The inhibitory effect on cytokine production relates to the substitution pattern. Whereas methylation of the two hydroxyl groups at C3 and C5 (pterostilbene, Figure 1D) gave only a slightly lower inhibition than resveratrol, methylation of the hydroxy groups at C3 and C4′ (pinostilbene, Figure 1E) resulted in a substantially weakened inhibition of cytokine release (40 µM pinostilbene gave ≈ 50% inhibition of IL-12 production compared to ≈81% inhibition for 40 µM resveratrol).

### 2.2. Monomer Derivatives Show Potent IL-12 Inhibition but Weak Inhibition of IL-10 and TNF-α Production in bmDCs Stimulated with E. Coli Nissle 1917

Stimulation with increasing MOI (0.3–3) of the Gram-negative *E. coli* Nissle 1917 resulted in a dose-dependent increase for IL-10, a slight decrease for IL-12, and no effects on TNF-α production (Figure 2A). All monomers significantly inhibited the *E. coli*-induced production of IL-12 in a dose-dependent manner (Figure 2, panels B–F). Piceatannol was the most potent inhibitor of IL-12 production (≈82% at 40 µM, Figure 2C). Trimethoxy-resveratrol also displayed more potent inhibition of IL-12 than resveratrol (75% inhibition vs. ≈61% for resveratrol, both at 40 µM, Figure 2, panel F vs. panel B). Only resveratrol and piceatannol displayed significant inhibition of IL-10 production, but only at high concentrations (30 μM and 40 μM, Figure 2, panels B,C). Only the trimethylated analogue and resveratrol showed potent inhibition of the *E. coli*-induced TNF-α production (Figure 2, panels B,F).

### 2.3. Effect of Dimeric Stilbenoids on the Bacterially Induced Cytokine Response

The stilbenoid dimers dehydro-δ-viniferin and *trans*-δ-viniferin were used in the same molar concentration range of stilbenoid monomers (0–40 μM) to assess whether dimerization affected bacterially induced cytokine response. Dehydro-δ-viniferin enhanced *L. acidophilus*-induced IL-12 response in a dose-dependent manner, whereas the IL-10 response showed a dose-dependent inhibition, and TNF-α gave a non-significant increase (Figure 3A). Treatment with *trans*-δ-viniferin did not affect the *L. acidophilus*-induced cytokine response pattern (Figure 3B). Neither dehydro-δ-viniferin nor *trans*-δ-viniferin affected the *E. coli*-induced production of any of the examined cytokines (Figure 3, panels C,D).

### 2.4. Number and Position of Hydroxy Groups Affect Stilbenoids Inhibition of Intracellular ROS

The antioxidant effect of the stilbenoid structures in the bmDCs was investigated by measuring their influence on the microbially induced intracellular ROS formation, as determined by oxidation of 6-carboxy-2′,7′-dichlorodihydrofluorescein diacetate (carboxy-H_2_DCFDA).

LPS-induced ROS formation was diminished for cells treated with the stilbenoid monomers resveratrol, piceatannol, and, to a lesser extent, pterostilbene (Figure 4A,C). Cells treated with pinostilbene or trimethoxy-resveratrol did not show a significant decrease of ROS formation, whereas dehydro-δ-viniferin and *trans*-δ-viniferin decreased LPS-induced ROS formation (Figure 4B,D) and dehydro-δ-viniferin led to slightly greater inhibition of ROS formation than *trans-*δ-viniferin.

### 2.5. Determination of the Total Antioxidant Capacity of Resveratrol and Resveratrol Derivatives

The antioxidant capacity of the monomeric and dimeric resveratrol derivatives was further examined by determining their total antioxidant capacity (TAC), as measured with both the 2,2′-azino-bis(3-ethylbenzothiazoline-6-sulfonic acid (ABTS) and the 2,2-diphenyl-1-picrylhydrazyl (DPPH) assay.

In agreement with data on intracellular ROS formation, piceatannol and resveratrol showed the highest TAC rates in ABTS assays. Pterostilbene’s TAC was lower than that of pinostilbene. As expected, trimethoxy resveratrol displayed no detectable TAC. Indeed, a significant correlation (*r* = 0.9339; R^2^ = 0.8722; *p* < 0.05) was observed between the number of hydroxyl groups present in the chemical structure of each monomer and their TAC, as measured by the ABTS assay. As for stilbenoid dimers, the highest TAC was observed for *trans*-δ-viniferin, whereas the TAC of dehydro-δ-viniferin was comparable to that of pinostilbene (Figure 5A). Again, a strong positive correlation was evident between the number of hydroxy groups in all stilbenoids (monomers and dimers) and the TAC evaluated by the ABTS assay (*r* = 0.7776; R^2^ = 0.6047; *p* < 0.05).

The DPPH assay confirmed that piceatannol and resveratrol had the highest TAC rates and trimethoxy-resveratrol the lowest. However, when using DPPH, pterostilbene displayed a higher TAC than pinostilbene, with values for pterostilbene similar to those measured for dehydro-δ-viniferin and *trans*-δ-viniferin (Figure 5B). By using the DPPH assay, a significant correlation (*r* = 0.9371; R^2^ = 0.8781; *p* < 0.05) was observed between TAC and the number of hydroxy groups present in the chemical structure of monomers.

### 2.6. Resveratrol and Dehydro-δ-Viniferin Affect Map Kinases Divergently

The striking difference in *L. acidophilus* NCFM-induced IL-12 production elicited by resveratrol and dehydro-δ-viniferin prompted us to investigate the mechanisms underlying the divergent activities of the two stilbenoids. Accordingly, it has been investigated how inhibitors of the Jun N-terminal kinase (JNK) and p38 affected the IL-12 and IL-10 responses to stimulation with *L. acidophilus* NCFM and how *L. acidophilus* (used alone or after a resveratrol or dehydro-δ-viniferin pretreatment) affected the expression of dual specificity phosphatase 1 (*Dusp1*) (Figure 6). While JNK inhibition gave a potent inhibition of IL-12 (73%) and IL-10 (96%) production, inhibition of p38 inhibition had only minor effects on IL-12 (11% inhibition), but strongly impaired IL-10 production (76% inhibition, Figure 6A). Thus, JNK activation seems indispensable for *L. acidophilus* NCFM-induced effects on IL-12 production.

To investigate whether resveratrol and dehydro-δ-viniferin affect the induction of the *Dusp1* expression, bmDCs were incubated for 30 min, with or without resveratrol (30 µM) or dehydro-δ-viniferin (30 µM), prior to stimulation with *L. acidophilus* NCFM (MOI 1). At multiple time points, cells were lysed and RNA extracted. bmDCs stimulated with *L. acidophilus* NCFM displayed increased *Dusp1* expression after 6 h, with a further increase at 8 h (Figure 6B). For cells incubated with resveratrol, the *Dusp1* expression showed a threefold increase at 4 h and then dropped gradually at 6 and 8 h. Cells incubated with dehydro-δ-viniferin showed only a very weak increase in *Dusp1* expression at 6 h, after which it dropped to a background level at 8 h (Figure 6B). The cytokine response in the supernatant from the stilbenoid-treated cells was also examined (Figure 6C). At 8 h after *L. acidophilus* NCFM stimulation, the IL-12 concentration in the supernatants of cells treated with dehydro-δ-viniferin increased more compared to untreated cells, and for resveratrol-treated cells, the IL-12 concentration remained close to zero for 8 h, then a subtle increase appeared. The production of IL-10 was almost zero for 8 h, with a drastic increase from 8 to 20 h. Treatment with the both stilbenoids led to a decreased IL-10 production.

## 3. Discussion

Resveratrol and its natural derivatives are regarded as anti-oxidative and anti-inflammatory compounds, but how their individual structures affect these activities is much less clear. Here, monomeric and dimeric derivatives of resveratrol have been used to scrutinize the effect of each single molecule by comparing their impact on the oxidative status of microbially stimulated dendritic cells, as well as on bacterially induced cytokine production in the same cell system. This latter approach also allowed us to take into appropriate consideration the distinct signaling pathways and cytokine production involved in the response of bmDCs towards different bacteria [10].

The disparate effects of stilbenoids on the cytokine production induced by the two bacterial strains have not been demonstrated before, but may be expected when considering the different signaling pathways induced by the two bacteria [6]. The Gram-negative *E. coli* Nissle 1917 induces pathways involving MyD88, as well as TIR domain-containing adaptor-inducing interferon-β (TRIF)-induced TLR signaling, depending on whether stimulation takes place from the plasma or the endosomal membrane. In contrast, the Gram-positive *L. acidophilus* NCFM exclusively stimulates from the endosomal membrane and only through MyD88 [6], in a process strictly dependent on endosomal degradation [11]. These differences cause the activation of diverse signaling pathways employing various signaling molecules and acting to a different extent and with different kinetics [12,13].

Kinases at large, and in the particular mitogen-activated protein (MAP) kinases, represent a group of signaling molecules that reportedly are inhibited by stilbenoids and other polyphenols [11,14,15], but bacteria exploit different activation patterns for kinases. For example, it was previously demonstrated that IL-12 induction by the Gram-positive bacteria *L. acidophilus* and *S*. *aureus* is strictly dependent on JNK and less on p38 [16,17]. In contrast, for Gram-negative bacteria *E. coli,* signaling (through TRIF) is dependent on p38 signaling [18]. In addition, the rate of endosomal degradation of the bacteria may affect the relative contribution of different MAP kinases. Indeed, recent reports have indicated that the kinetics of p38 activation involved in the signaling to some Gram-positive bacteria depends on how readily bacteria are degraded in endosomes [16,19]. Hence, the effect of specific stilbenoids may depend on their ability to bind to one or more MAP-kinases, as well on which kinases are most relevant to cytokine production.

This work also reports high variation of cytokine production in bmDCs depending on the stilbenoid’s structure. The most striking variation was the diverging effect of resveratrol and the resveratrol dimer, dehydro-δ-viniferin, on *L. acidophilus* NCFM-induced IL-12 production. Whereas resveratrol, similarly to other stilbenoids, inhibited the *L. acidophilus* NCFM-induced production of IL-12, dehydro-δ-viniferin enhanced it. *L. acidophilus* NCFM-induced IL-12 is strictly dependent on the MAP kinase JNK [17] and an early transcription of the p38-induced *Dusp1* was reported to lead to a decrease in IL-12 production [16]. The results reported here indicate that dehydro-δ-viniferin almost completely suppresses *L. acidophilus* NCFM-induced *Dusp1* expression, whereas resveratrol leads to an earlier and stronger *Dusp1* expression. As DUSP1 inactivates both JNK and p38 [20], inhibition of *Dusp1* expression by dehydro-δ-viniferin indicates that p38 activation is abrogated, resulting in persistent phosphorylation and activity of JNK. Conversely, the earlier enhanced induction of *Dusp1* expression and the almost full abrogation of IL-12 during the first 8 h in the presence of resveratrol could indicate that JNK was inhibited. Here, it is relevant that some of the kinase signaling pathways may activate both JNK and p38, and that their actions may be redundant, at contrast with other activation pathways that are highly specific and involve only one of the two MAP kinases [20,21].

The cytokine-modulating capacity of each stilbenoid may be influenced by several properties of the molecule. Firstly, the capacity of the stilbenoid to pass through the plasma membrane may affect the proportion of the stilbenoid that actually reaches intracellular targets. Methylated monomers, as well as the two dimers, more readily pass the hydrophobic membrane due to their higher lipophilicity compared to the parent compounds. In support of this, Dias et al. [22] demonstrated a 40 times higher serum concentration of trimethoxy-resveratrol than resveratrol in mice after oral administration. The documented differences in pharmacokinetics have been suggested to account for the higher biological activity of pterostilbene and other methylated compounds over the parental compound resveratrol [23]. Given the comparable effects of resveratrol and trimethoxy-resveratrol in *E. coli*-stimulated bmDCs, the differences between these stilbenoids observed in vivo seem more likely to be caused by the transformation of resveratrol into other molecules. As a matter of fact, resveratrol was demonstrated to be transformed (e.g., by oxidation by the gut commensals [24]) in the digestive tract.

When investigating the capacity of each stilbenoid to affect LPS-induced ROS formation in the dendritic cells, methylated monomers were found to be much less efficient than resveratrol, as expected. Surprisingly, the stilbenoid dimers showed weak ROS-inhibiting activity in the LPS-stimulated cells and also showed a reduced antioxidant capacity when TAC was determined by DPPH assay. This suggests that the lower antioxidant capacity of the dimers is related to their lower anti-radical activity rather than to poor cellular uptake or to molecular interaction with intracellular targets involved in the activation of ROS formation, e.g., nuclear factor erythroid-derived 2-like pathway [25,26,27], contributing to the final effect on counteracting ROS formation [28].

Although the strength of the antioxidant effect of polyphenols has been described and depends on their structural properties, such as the Bors criteria [29], this study makes evident a linear correlation between TAC and the number of hydroxyl groups in stilbenoids, especially for their monomeric forms. This is in line with a recent paper by Platzer et al., who studied several phenolic compounds and found that the number of hydroxyl groups influenced the antioxidant activity more accurately than the Bors criteria [30]. Although the ABTS and DPPH assays are both single electron transfer-based assays and determined the anti-radical capacity of the molecules in the sample, slight differences were found between the two methods. These discrepancies may be due to the stilbenoids’ solubility in the reaction media (water and hydro-methanolic solution in ABTS and DPPH assay, respectively) and electron transfer kinetic issues [31,32], compared to the ROS formation measured in LPS-stimulated dendritic cells.

Endosomal ROS formation is required for efficient degradation of some phagocytosed bacteria and is often, but not always, required to release ligands for the TLRs and, thus, for cytokine production. This makes it relevant to assess if there is a relationship between ROS formation, the requirement of bacterial degradation, and cytokine production. Efficient endosomal degradation of *L. acidophilus* NCFM is a prerequisite for cytokine production, in particular for IL-12 [6,11], as it is for many other Gram-positive bacteria [33,34]. In contrast, many Gram-negative bacteria, such as *E. coli* Nissle 1917, shed TLR-stimulating LPS from the surface and are, therefore, less dependent on endosomal degradation to induce cytokine production [35,36]. The stilbenoids affected the cytokine production induced by *L. acidophilus* and *E. coli* differently with the most pronounced effects on the *L. acidophilus*-induced cytokine production. Resveratrol and piceatannol exhibited the most potent inhibitory effects in *L. acidophilus*-induced cytokines; a part of the actions of these two stilbenoids might be due to an inhibition of the ROS formation. This is, however, purely speculative.

Interestingly, in contrast to the monomeric stilbenoids, the dimeric stilbenoid dehydro-δ-viniferin enhanced the IL-12 production induced by *L. acidophilus*. Notably, IL-12 treatment of WT mice before pulmonary challenge with MRSA protected mice against bacterial growth and increased survival [37].

The IL-12-enhancing capacity reported here has not been shown, to our best knowledge, for any stilbenoids or for other polyphenols. Considering the rather potent antimicrobial activity towards Gram-positive pathogens, dehydro-δ-viniferin may represent a dual-pronged strategy for the treatment of infections by Gram-positive bacteria such as Staphylococcus aureus, by acting directly on the pathogen as well as indirectly, through an enhanced cellular immune response against the pathogen.

## 4. Materials and Methods

### 4.1. Generation of Murine Bone Marrow-Derived Dendritic Cells (bmDCs)

The bmDCs were generated as described previously [4,38]. In brief, the femur and tibia were removed from C57BL/6NTac mice (Taconic, Lille Skensved, Denmark). The bone marrow cells were cultivated at 3 × 10^5^ cells/mL in RPMI medium containing 10% fetal calf serum and 15 ng/mL granulocyte-macrophage colony-stimulating factor (GM-CSF). After eight days of incubation, non-adherent bmDCs were used. All animals used as sources of bone marrow cells were housed under conditions approved by the Danish Animal Experiments Inspectorate (Forsøgdyrstilsynet) according to The Danish Animal Experimentation Act; LBK no. 474 from 15 May 2014, and experiments were carried out in accordance with the guidelines of ‘The Council of Europe Convention European Treaty Series (ETS)123 on the Protection of Vertebrate Animals Used for Experimental and Other Scientific Purposes’.

### 4.2. Bacterial Strains and Stilbenoid Synthesis

The Gram-positive bacterium *L. acidophilus* NCFM (Danisco, Copenhagen, Denmark) was grown anaerobically overnight at 37 °C in de Man Rogosa Sharp broth (Merck, Darmstadt, Germany). The Gram-negative bacterium *E. coli* Nissle 1917 O6:K5:H1 (Statens Serum Institute, Copenhagen, Denmark) was grown aerobically overnight at 37 °C in Luria-Bertani broth. The bacteria were subcultured twice, harvested by centrifugation at 1250× *g* for 10 min, and washed twice in sterile PBS. The bacteria were seeded in Petri dishes, killed with UV pulsation, and the bacterial viability was verified. Stilbenoids (Figure 7) were synthesized as previously described [9], solubilized in DMSO to a concentration in excess of 10 mM (6.38 mM for *trans*-δ-viniferin), and further diluted in PBS or growth media as appropriate. Final DMSO concentrations below the 0.5% (*v*/*v*) thresholds did not lead to significant changes of the cytokine response in the system used in this study.

### 4.3. Assessment of Bacteria-Induced Cytokine Response

For determination of the appropriate MOI to be used in further tests, bmDCs (2 × 10^6^ cells/mL) were stimulated with increasing concentration (MOI 0.3–3) of *L. acidophilus* NCFM or *E. coli* Nissle 1917 and incubated for 20 h at 37 °C, 5% CO_2_ before the supernatant was harvested. For examinations of the stilbenoids’ effect on the cytokine response, bmDCs (2 × 10^6^ cells/mL) were incubated for 30 min at 37 °C, 5% CO_2_ in the presence of increasing concentrations (0–40 μM) of the individual stilbenoid, prior to stimulation with *L. acidophilus* NCFM (MOI 1) or *E. coli* Nissle 1917 (MOI 1). For assessing the effects of inhibition of MAP kinases, SP600125 (a JNK inhibitor, 25 µM) and SB203580 (a p38 inhibitor, 10 µM) were added to bmDCs and incubated for 30 min at 37 °C, 5% CO_2_ prior to stimulation with *L. acidophilus* NCFM. After 20 h of incubation, the culture supernatants were harvested and stored at −20 °C until analysis. The cytokine concentration in the culture supernatants was assessed using the Duoset™ ELISA kits for mouse IL-12p70 (DY419), IL-10 (DY417), and TNF-α (DY410). All kits were from R&D Systems^®^ (Minneapolis, MN, USA) and used according to the manufacturer’s instructions.

### 4.4. Assessment of Intracellular ROS Formation

Murine bmDCs (2 × 10^6^ cells/mL) were treated with 5 μM carboxy-H2DCFDA (Invitrogen™, Carlsbad, CA, USA) before the addition of resveratrol-derived stilbenoid monomers or dimers (30 μM, final concentration) and incubated for 30 min at 37 °C and 5% CO_2_. Samples were then stimulated with 100 ng/mL LPS (*E. coli* O26:B6, Sigma Aldrich, St Louis, MO, USA) and incubated for another 4 h at 37 °C and 5% CO_2_. Controls included samples without stilbenoids and LPS as well as samples with stilbenoids but without LPS and samples not treated with carboxy-H_2_DCFDA. Assessment of the intracellular ROS formation was carried out by measuring the intensity of the cellular fluorescence generated by oxidation of the internalized carboxy-H_2_DCFDA on a BD FACS Diva flow cytometer (BD Biosciences, Franklin Lakes, NJ, USA). All events counted (20,000/sample) were included in the analysis. Comparison of the intracellular ROS formation in various samples was based on the mean fluorescent intensity (MFI) of treated bmDCs. The flow cytometry results were analyzed using the FlowJo™ software (version 10.6.2, BD Life Sciences, Ashland, OR, USA).

### 4.5. In Vitro Total Antioxidant Capacity (TAC) Determination

TAC was assessed by the ABTS and DPPH assays [39] on the basis of the ability of the antioxidant molecules to reduce the radical cation of 2,2′-azino-bis-(3-ethylbenzothiazoline-6-sulfonic acid) (ABTS) and 2,2-diphenyl-1-picrylhydrazyl (DPPH). Ten microliters of a 1 mM stock solution of each species were added to 990 μL of 80μM ABTS^+^ and 100 μM·DPPH and the quenching of the absorbance at 734 nm for 1 min and at 517 nm for 30 min for ABTS^·+^ and DPPH·, respectively, was monitored. Values obtained for each sample were compared to the concentration–response curve of the standard Trolox solution and expressed as mM of Trolox equivalent (TE).

### 4.6. Assessment of Fold Change in Gene Expression in bmDC

Murine bmDCs (2 × 10^6^ cells/mL) were stimulated with *L. acidophilus* NCFM (MOI 1) and incubated at 37 °C and 5% CO_2_. To determine the effect of stilbenoids on the expression of specific genes in bmDCs, cells were incubated with 30 µM of the examined stilbenoids for 30 min at 37 °C and 5% CO_2_ prior to stimulation. Supernatant from stimulated bmDCs was removed and mRNA was extracted using MagMAX-96 Total RNA Isolation Kit (AM1830, Applied Biosystems, Foster City, CA, USA). The extracted RNA (500 ng for each sample) was converted to cDNA using Applied Biosystems™ High-Capacity cDNA Reverse Transcription Kit. Applied Biosystems™ TaqMan™ qPCR with a specific TaqMan MGB probe and primer sequences were then used to estimate the expression of *Dusp1* (Mm00457274_g1). *Actb* (Mm00607939_s1) was used as the reference gene to define ΔCt for each sample (ΔCt_target_ = Ct_target_ − Ct_reference_). Fold change expression was estimated using the ΔΔCt method, where first the ΔCt of the blank (unstimulated cells) from each time point is subtracted from the ΔCt of a target sample (stimulated cells) at each time point (ΔΔCt = ΔCt_target_ − ΔCt_control_), with fold change in gene expression being defined by 2^(−(ΔΔCt)^.

### 4.7. Statistical Analysis

Statistical analysis was performed by using the GraphPad Prism software (version 9.3.1, GraphPad Software, San Diego, CA, USA). Unless otherwise specified, results are illustrated as means ± SD. The results were analyzed by one-way analysis of variance (ANOVA) to determine significance. *p* values less than 0.05 were considered statistically significant and indicated by asterisks. * *p* ≤ 0.05, ** *p* ≤ 0.01, *** *p* ≤ 0.001, **** *p* ≤ 0.0001.

## 5. Conclusions

This work demonstrates that the potential of stilbenoids to affect the cytokine production induced by bacteria in murine dendritic cells depends on the specific structure of the stilbenoid, as well as on the type of microbial stimulant. Specific knowledge of stilbenoid molecular determinants modulating the cytokine response of dendritic cells, i.e., the cells orchestrating the type of immune response against pathogens, may be exploited in outlining future strategies for development of new drugs and nutraceuticals.

## Figures and Tables

**Figure 1 ijms-24-02731-f001:**
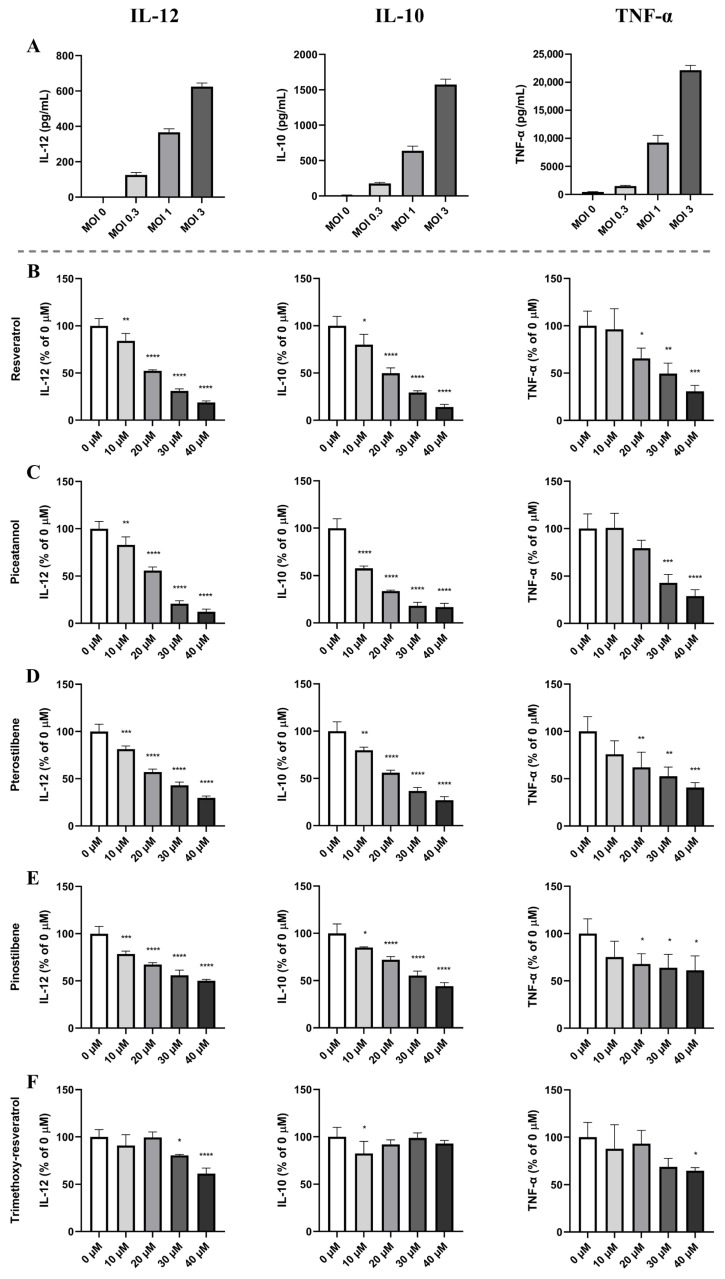
Resveratrol and piceatannol hold vigorous cytokine-inhibiting roles in *L. acidophilus* NCFM-stimulated bmDCs, and methylation of resveratrol weakens the inhibitory capabilities. Murine bmDCs were stimulated with *L. acidophilus* NCFM at multiplicity of infection (MOI) at rates of 0.3, 1, and 3 (**A**) and incubated for 20 h. To assess the effect of monomers on the cytokine response, bmDCs were incubated with increasing concentrations (0–40 µM) of resveratrol (**B**), piceatannol (**C**), pterostilbene (**D**), pinostilbene (**E**), and trimethoxy-resveratrol (**F**) for 30 min before *L. acidophilus* NCFM (MOI 1) was added to the samples. Cytokine levels in the supernatant for stilbenoid-treated samples (**B**–**F**) are shown as percentages of samples with no stilbenoid treatment (0 µM). The depicted data are representative of at least three experiments. * *p* ≤ 0.05, ** *p* ≤ 0.01, *** *p* ≤ 0.001, **** *p* ≤ 0.0001.

**Figure 2 ijms-24-02731-f002:**
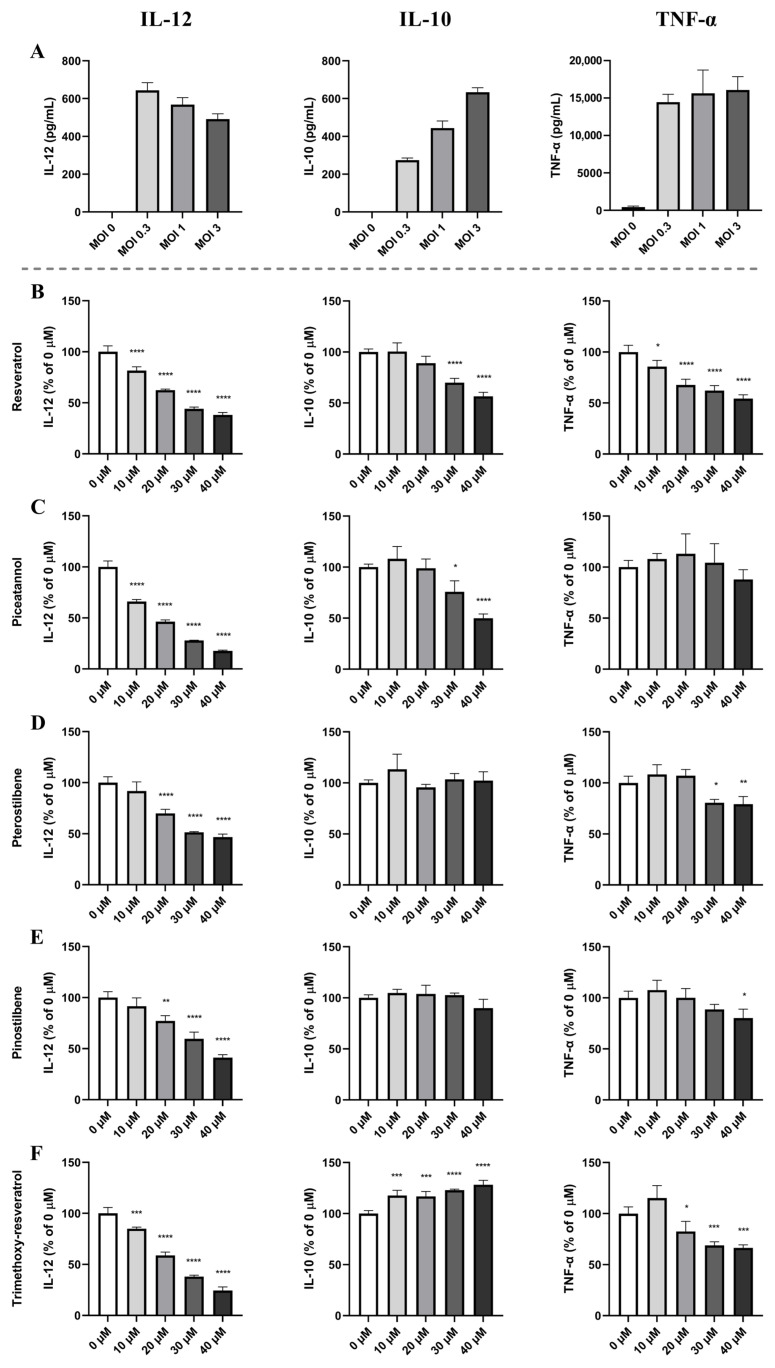
Methylation of resveratrol weakens the IL-10 inhibitor activity in *E. coli* Nissle 1917-stimulated bmDCs. Murine bmDCs were stimulated with *E. coli* Nissle 1917 at multiplicity of infection (MOI) rates of 0.3, 1, and 3 (**A**) and incubated for 20 h. To assess the effect of monomers on the cytokine response, bmDCs were incubated with increasing concentrations (0–40 µM) of resveratrol (**B**), piceatannol (**C**), pterostilbene (**D**), pinostilbene (**E**), and trimethoxy-resveratrol (**F**) for 30 min before *E. coli* Nissle 1917 (MOI 1) was added to the samples. Cytokine levels in the supernatant for stilbenoid-treated samples (**B**–**F**) are shown as percentages of samples without stilbenoid treatment (0 µM). The depicted data are representative of at least three experiments. * *p* ≤ 0.05, ** *p* ≤ 0.01, *** *p* ≤ 0.001, **** *p* ≤ 0.0001.

**Figure 3 ijms-24-02731-f003:**
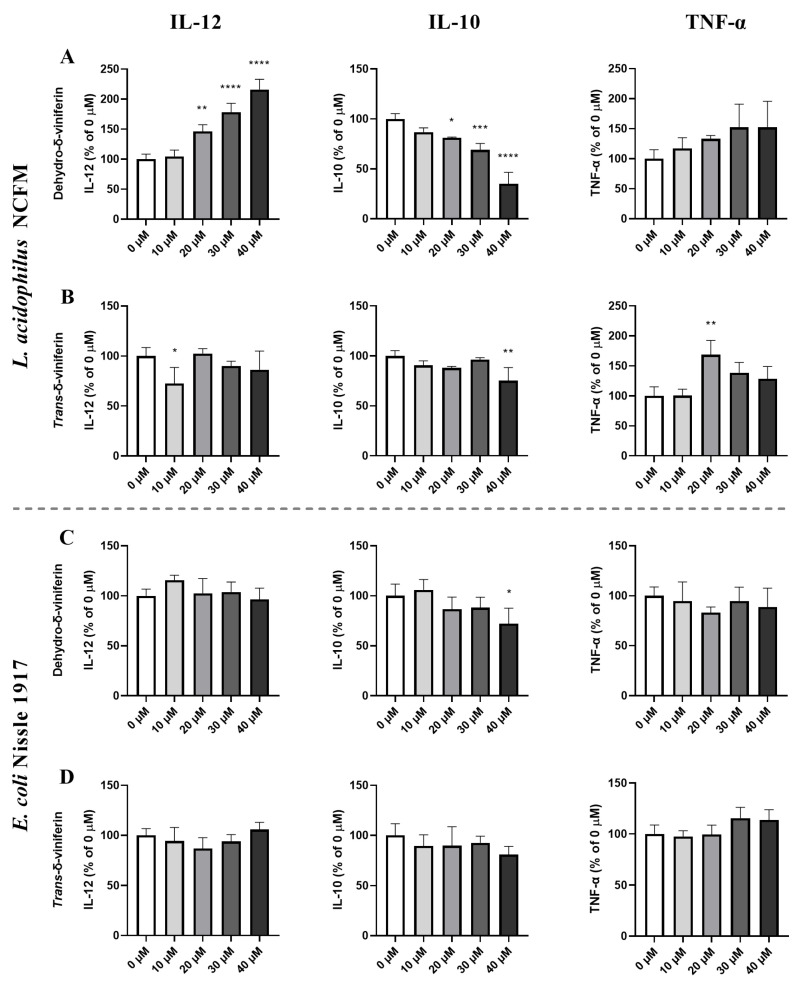
Effect of stilbenoid dimers on the bacterially induced cytokine response depends on the bacterial strain. Murine bmDCs were incubated with increasing concentrations (0–40 µM) of dehydro-δ-viniferin (**A**,**C**) and *trans*-δ-viniferin (**B**,**D**) for 30 min before stimulation with *L. acidophilus* NCFM (multiplicity of infection (MOI) 1) (**A**,**B**) or *E. coli* Nissle 1917 (MOI 1, **C**,**D**) and incubated for 20 h. Cytokine levels in the supernatant for the stilbenoid treated samples are shown as percentages of samples without stilbenoid treatment (0 µM). * *p* ≤ 0.05, ** *p* ≤ 0.01, *** *p* ≤ 0.001, **** *p* ≤ 0.0001.

**Figure 4 ijms-24-02731-f004:**
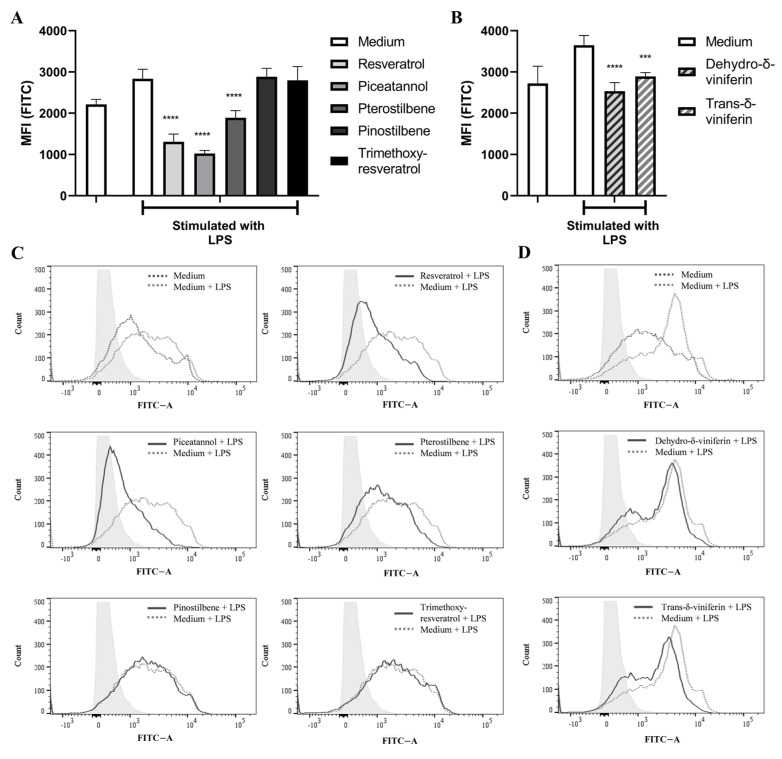
Effect of stilbenoid monomers and dimers on LPS-induced ROS formation in bmDCs. Murine bmDCs were administered carboxy-H_2_DCFDA and incubated in the absence or in the presence of the given stilbenoids (30 µM) for 30 min before stimulation with LPS (100 ng/mL). After 4 h, ROS formation in bmDCs was determined by using flow cytometry to quantitate formation of oxidized carboxy-H_2_DCFDA. The mean fluorescence index (MFI) is shown in panels (**A**) (monomers) and (**B**) (dimers). The original flow cytometry results for monomers and dimers are grouped in (**C**,**D**), respectively. *** *p* ≤ 0.001, **** *p* ≤ 0.0001.

**Figure 5 ijms-24-02731-f005:**
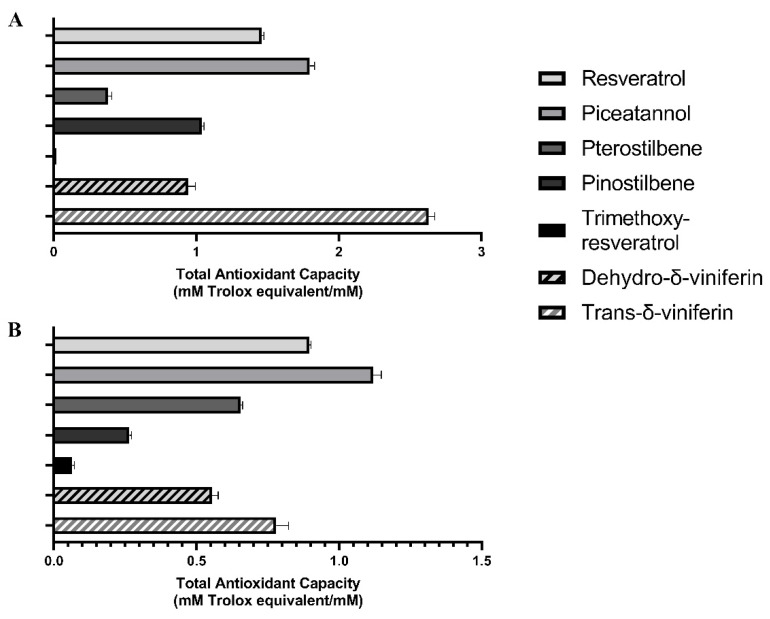
Total antioxidant capacity (TAC) in stilbenoid monomers and dimers. The TAC rates of the five monomeric stilbenoids and the two dimeric stilbenoids were determined using ABTS (**A**) and DPPH assay (**B**). TAC values were calculated for stilbenoids at an equimolar concentration (1 mM) and are expressed as mM Trolox equivalent/mM stilbenoids.

**Figure 6 ijms-24-02731-f006:**
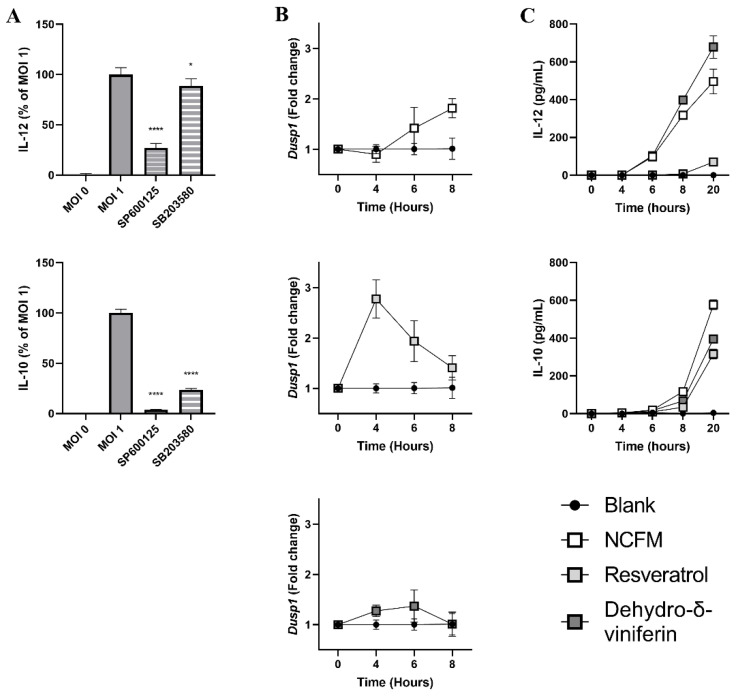
Resveratrol and dehydro-δ-viniferin interfere differently with MAP kinase pathways. Murine bmDCs were treated with or without SP600125 (JNK-inhibitor, 25 µM) or SB203580 (p38-inhibitor, 10 µM) for 30 min before stimulation with *L. acidophilus* NCFM (MOI 1) and incubated for 20 h (**A**). Cytokine levels in the supernatant for stilbenoid-treated samples are shown as percentages of samples without stilbenoid treatment (MOI 1). The expression rates of *Dusp1* in bmDCs (0 to 8 h) upon stimulation with *L. acidophilus* NCFM with or without 30 min pre-incubation with resveratrol (30 µM) or dehydro-δ-viniferin (30 µM) are shown (**B**). *Dusp1* expression is shown as fold change relative to unstimulated cells (black) with *Actb* as the reference gene. The cytokine responses in the supernatants of bmDCs stimulated with *L. acidophilus* NCFM over time (white) with or without resveratrol (30 µM) (light grey) or dehydro-δ-viniferin (30 µM) (dark grey) added 30 min prior to stimulation are shown (**C**). * *p* ≤ 0.05, **** *p* ≤ 0.0001.

**Figure 7 ijms-24-02731-f007:**
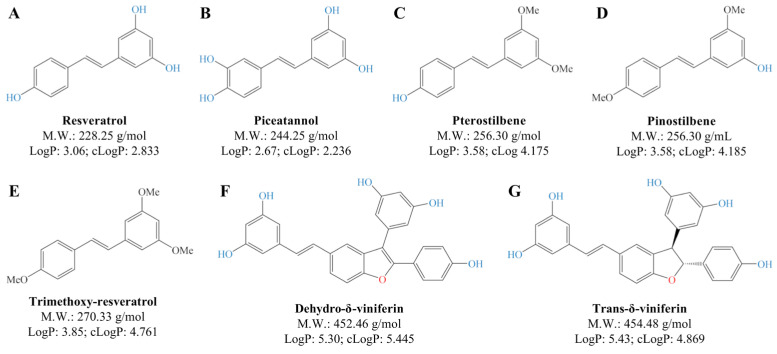
Chemical structures of the utilized stilbenoid monomers and dimers. Overview of the examined stilbenoids, listing the structure, molecular weight, LogP values, and cLogP values for resveratrol (**A**), piceatannol (**B**), pterostilbene (**C**), pinostilbene (**D**), trimethoxy-resveratrol (**E**), dehydro-δ-viniferin (**F**), and *trans*-δ-viniferin (**G**).

## Data Availability

Not applicable.

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
