# Peer review of "Investigation of the Effects of Monomeric and Dimeric Stilbenoids on Bacteria-Induced Cytokines and LPS-Induced ROS Formation in Bone Marrow-Derived Dendritic Cells"

_ijms, 2023, doi:10.3390/ijms24032731_

Round 1

Reviewer 1 Report (New Reviewer)

Johnsen and colleagues described the anti-inflammatory and antioxidant effects of resveratrol and its derivatives in murine bone marrow-derived dendritic cells (bmDC). The study sounds interesting and provides novel data on the antioxidant and anti-inflammatory effect of resveratrol derivatives. The major limitations of the present study are the unique non-human cell model used, and that the anti-inflammatory effect of resveratrol analogues was tested only by quantifying a small number of cytokines (TNFalpha, IL-12 and IL-10). Despite these limitations, the authors clearly described an interesting linear correlation between the antioxidant capability of resveratrol derivatives and the number of hydroxy groups in these compounds, in particular in monomers. In addition, methylation of hydroxy groups led to reduced anti-inflammatory capability of analogues. This study may help to understand how to develop resveratrol-derived antioxidants with improved efficacy. However, the following major concerns arisen from the current version of the manuscript:

11. The authors did not state neither in the title nor in the abstract that bmDC were obtained from a murine model. It should be clearly stated at least in the abstract

22. The authors show a different pro-inflammatory activity sustained by E. coli compared with L. acidophilus. A possible explanation may partially reside in the different cytotoxicity induced by different bacteria. The authors should provide data indicating that the two bacterial strains used in this study do not promote remarkable cell death in bmDC within 20h of treatment.

33. Authors show that all monomers inhibited E.coli-dependent IL-12 release, whereas only trimethylresveratrol (TMR) and resveratrol reduced TNF-alpha release, and only resveratrol and piceatannol displayed a significant inhibition of IL-10 release in bmDC. Lack of anti-bacterial effect against E. coli (and L. acidophilus) by resveratrol and its derivatives should be provided in order to exclude that the different inhibition of cytokine release promoted by some compounds in bmDC is associated with antibacterial properties rather than anti-inflammatory capabilities.

44. Authors show that LPS-induced ROS formation in bmDC is inhibited by resveratrol, piceatannol, and pterostilbene, whereas no effect was observed using pinostilbene and TMR. On the contrary, data obtained in bmDC stimulated with the Gram negative E. Coli indicated that treatment with piceatannol and pterostilbene did not change both IL-10 and TNF release in bmDC, whereas TMR showed interesting inhibitory effect on TNF. How the authors justify this discrepancy of stilbenoid-mediated effect upon LPS and Gram negative challenges? Is the cytokine release independent from ROS production? It should be further discussed in the text.

Minors:

55. In Figure 2 authors show the effect of resveratrol monomeric derivatives on cytokine release in bmDC. It is not clear which type of control has been used as “vehicle” in each panel. Did the authors use the same solvent (i.e. DMSO or ethanol) to solubilize resveratrol and all the analogues? It should be clearly indicated in figure legend the type of vehicle and reported in each graph y axis as (% of “vehicle”). Vehicles should be indicated also in methods.

66. Figures 2, 3 and 4 should indicate which kind of stilbenoid was used next to each panel group, in order to facilitate the readers.

77. Figure 5, panel C, is almost unreadable. Authors should enlarge the panel and the font size of graph axis.

88. In discussion, TRIF should be written in full whenever it appears in the text (line 269) and left abbreviated in line 283.

99. In the Conclusion session, it should be stated that authors “…have demonstrated that the potential of stilbenoids to affect the cytokine production induced by bacteria in murine dendritic cells depends on the specific structure of stilbenoids…”.

Author Response

Reviewer 2 Report (New Reviewer)

Overall, the paper by Johnsen et al. is interesting and properly designed. My main concern is about the text manuscript, which should be carefully checked and improved.

INTRODUCTION

Line 51: please, change “we have previously…” with “it has been previously…”. Accordingly, modify the text throughout the entire manuscript.

Lines 53-55: this sentence should be deleted from the introduction paragraph. Alternatively, please provide a suitable reference.

Lines 61-66: this sentence is too long and sounds hardly understandable. It should be rewritten.

Lines 68-74: Authors should clearly state the aims of the present study. Figure 1 should be moved to the “materials and methods” section.

RESULTS

The text sounds somewhat redundant. Duplicates of “materials and methods” (e.g., lines 85-86, 90-93 etc.) should be deleted.

Some terms appear more suitable in the discussion paragraph (e.g., “of note” at line 97, …”depended on…” at line 98, “In contrast to…” at line 115 and 126, lines 128-131, lines 151-152, etc.).

Lines 207-240: most of these sentences are unsuitably included in the results’ paragraph and should be moved elsewhere (introduction, materials and methods, discussion).

DISCUSSION

Line 254: herein, as well as throughout the entire manuscript, please avoid the form “we …”, using the third person format.

Line 275: please, add a suitable reference.

Overall, this paragraph should be shortened, and the text format improved.

MATERIALS AND METHODS

Lines 397-399: please, add further technical details.

Lines 405-416: please, add further details about control samples included in each run.

Lines 417-421: further technical details should be provided.

Round 2

Reviewer 1 Report (New Reviewer)

The authors sufficiently improved the manuscript. No further concerns from my point of view.

This manuscript is a resubmission of an earlier submission. The following is a list of the peer review reports and author responses from that submission.

Round 1

Reviewer 1 Report

Comparing the effects of monomeric and dimeric stilbenoids 2 on bacteria-induced cytokines and LPS-induced ROS formation 3 in bone marrow-derived dendritic cells

Stilbenoids, as a group of natural substances against bacterial pathogens, are promising targets for investigation. They can be considered as potential naturally derived drugs, which are preferred over synthetic compounds in pharmaceutical markets.

In my opinion the structure of the reviewed article is well-thought-out and clear. The methods used are appropriate. Each step of the investigation is explained very well. I have few issues for the Authors to improve.

In the Introduction the authors highlighted the aims, significance and the novelty of their work. It is well written and comprehensive.

When it comes to the Material and Methods section I think that including the flow cytometry gating strategy would add some value to the manuscript. Do the Authors include cells not stimulated with LPS as gating control?

Figure 5 has unclear caption. The graph B hasn’t been cited in the caption.The bottom flow cytometry graphs in Fig.5 are extremely blurred– please correct this.

Please improve also the quality of the Figure 1.

The Discussion definitely meets the requirements of the professional scientific article. The authors have appropriately discussed the limitations of the methods used. The conclusions are supported by the presented data.

Author Response

Stilbenoids, as a group of natural substances against bacterial pathogens, are promising targets for investigation. They can be considered as potential naturally derived drugs, which are preferred over synthetic compounds in pharmaceutical markets.

In my opinion the structure of the reviewed article is well-thought-out and clear. The methods used are appropriate. Each step of the investigation is explained very well. I have few issues for the Authors to improve.

In the Introduction the authors highlighted the aims, significance and the novelty of their work. It is well written and comprehensive.

We sincerely thank the Reviewer for her/his appreciation of our work and we are convinced that thanks to her/his suggestions the quality of the manuscript has significantly improved.

When it comes to the Material and Methods section I think that including the flow cytometry gating strategy would add some value to the manuscript. Do the Authors include cells not stimulated with LPS as gating control?

We did include non-stimulated cells in the experiment. These are depicted in figure 5c, upper left graph. However, we acknowledge that the graph was not clearly explained and have sought to amend this in an improved version of the figure.

Regarding the gating strategy: We included all cells, i.e. we did not exclude dead cell debris or cells appearing as doublets. This means that we compared the MFI for all cells in each sample. This has now been specified (l.356-357).

Figure 5 has unclear caption. The graph B hasn’t been cited in the caption. The bottom flow cytometry graphs in Fig.5 are extremely blurred– please correct this.

The upper figures were wrongly cited as (A and C) in the caption. This has now been corrected to (A and B). We thank the reviewer for pointing out this error.

The lower part of figure 5 has been changed to a version of higher quality.

Please improve also the quality of the Figure 1.

Thanks for the indication. In the revised version the quality of the figure 1 has been improved.

The Discussion definitely meets the requirements of the professional scientific article. The authors have appropriately discussed the limitations of the methods used. The conclusions are supported by the presented data.

Thanks for the kind appreciations.

Reviewer 2 Report

In the study presented by Johnsen et al. the authors compare five monomeric and two dimeric stilbenoids for their capability to modulate the production of bacterially-induced cytokines (IL-12, IL-10, and TNFα) as well as LPS-induced ROS in mouse BM-derived DC. 

The manuscript is composed of two separate parts. In the first part the effect of stilbenoids on the production of just three cytokines generated by BM-derived DC in response to the gram-positive bacterium L. acidophilus NCFM and gram-negative bacterium E. coliNissle were addressed as a potential anti-inflammatory activity. In the second part the effect of stilbenoids on ROS formation upon LPS stimulation in BM-DC was analyzed.

The data appear accurately generated.

However, there are a number of concerns that militate against publication.

The major concern is that the data presented are merely descriptive without substantial novel mechanistic or conceptual insights.

The study lacks a hypothesis-driven concept for the analysis of stilbenoids on BM-DC. If the idea was to show that anti-inflammatory activity is based on the anti-oxidative activity of stilbenoids, then the very approach of first comparing a bacterial immune response of DC with LPS-induced ROS formation would not be useful.

The final conclusion in the abstract states: 

“In summary, the anti-inflammatory and the antioxidant activity showed no straight relationship, and appeared related to several factors, such as the type of the pro-inflammatory signal, and the chemical structure and bioavailability of the stilbenoids.”

None of the “several factors” including the pro-inflammatory signals or the chemical structure were clearly identified and bioavailability was not addressed. Thus, none of these conclusions were supported by the presented data. Since the two effects, the anti-inflammatory and anti-oxidative effects of stilbenoids, showed no straight relationship” and the aforementioned conceptual shortcomings, the study is of poor scientific value.

In summary, I cannot support publication of this manuscript in IJMS and have thus no further specific comments.

Minor point:

Since BM-DC were generated from mice the authors should follow the “Ethical Guidelines for the Use of Animals in Research” (see Instructions for Authors) and provide details in the Material and Methods section.

Author Response

In the study presented by Johnsen et al. the authors compare five monomeric and two dimeric stilbenoids for their capability to modulate the production of bacterially-induced cytokines (IL-12, IL-10, and TNFα) as well as LPS-induced ROS in mouse BM-derived DC.

The manuscript is composed of two separate parts. In the first part the effect of stilbenoids on the production of just three cytokines generated by BM-derived DC in response to the gram-positive bacterium L. acidophilus NCFM and gram-negative bacterium E. coliNissle were addressed as a potential anti-inflammatory activity. In the second part the effect of stilbenoids on ROS formation upon LPS stimulation in BM-DC was analyzed.

The data appear accurately generated.

We thank Reviewer for her/his suggestions. We carefully revised our manuscript according to the Reviewer’s recommendations, which we believe significantly improved the quality of the paper.

However, there are a number of concerns that militate against publication.

The major concern is that the data presented are merely descriptive without substantial novel mechanistic or conceptual insights.

We agree that the data are descriptive in nature and that we did not directly investigate the mechanisms behind the effect of the stilbenoids tested in this study. This was however not the aim of the current study (please see next point), which we believe, nevertheless holds substantial novelty:

  1. A comparison of the effect of different stilbenoids in antigen presenting cells stimulated with a gram positive and a gram negative bacteria, respectively, has not been done before and clearly showed that the immune modulating effect of the stilbenoids depends on both specific structural features of tested compounds and the stimulating bacteria.
  2. The effect of the dimeric stilbenoid dehydro-δ–viniferin on the gram-positive bacteria stimulated DCs is surprising and certainly presents interesting perspectives that may be exploited in drug development. In fact, in contrast to the monomeric stilbenoids, the dimeric stilbenoid dehydro-δ-viniferin enhanced the IL-12 production induced by L. acidophilus. Considering that dehydro-δ-viniferin is endowed with significant antimicrobial activity, being 200 times more active than monomers (10.1038/s41598-019-55975-1), it could be considered a difunctional compound, acting directly on the pathogen as well as indirectly, through an enhanced cellular immune response against the pathogen.
  3. Our idea was to evaluate if the stilbenoids’ immune-modulating effect was solely related to their antioxidant activity or to multiple actions on the cytokine production pathways. Cytokine induction (in particular IL-12) upon stimulation of DCs with gram-positive bacteria often requires endocytosis and endosomal degradation to release TLR-ligands from the bacteria. Accordingly, ROS production may be imperative for the endocytic degradation and thus for TLR-stimulation. On this basis, depending on the bacterial stimuli, a reduction of the ROS by antioxidants would hamper bacterial degradation and thus cytokine production.

Our data indicate that there is no straight relationship between the immune modulating effect of the stilbenoids and their antioxidant activity. All monomers except trimethoxy-resveratrol inhibited L. acidophilus NCFM induced IL-12, IL-10, and TNF-α production, while dehydro-δ-viniferin increased the IL-12 production. Despite having moderate to high total antioxidant activity, the two dimeric stilbenoids were weak inhibitors of LPS-induced ROS formation. However, as remarked in the discussion we see a correlation between the ROS inhibition by the monomeric stilbenes and the inhibition of IL-12 in the gram-positive stimulated bmDCs. Whether this reflects the inhibited ROS production or effects on signalling molecules is however not elucidated.

Hence, even though we did not address specific mechanisms of action of the investigated stilbenoids, the study does provide substantial new insights regarding the different effects of the selected stilbenoids depending on the three points described above.

We improved the discussion part to properly address the criticisms raised by the reviewer.

The study lacks a hypothesis-driven concept for the analysis of stilbenoids on BM-DC. If the idea was to show that anti-inflammatory activity is based on the anti-oxidative activity of stilbenoids, then the very approach of first comparing a bacterial immune response of DC with LPS-induced ROS formation would not be useful.

The idea was not to show that an anti-inflammatory activity of a stilbene is based on the anti- oxidative effect.

Our main hypothesis for the work presented in this manuscript was that the effect of the various stilbenoids may depend on the bacterial stimuli and thus stilbenoids may affect the type of immune response induced by the bacteria (e.g. abrogation of the IL-12 may inhibit development of a Th1 cells needed for an efficient elimination of the bacteria). This has now been clearly stated in the abstract.

Cytokine induction (in particular IL-12) upon stimulation of DCs with gram-positive bacteria often requires endocytosis and endosomal degradation to release TLR-ligands from the bacteria. Accordingly, ROS production may be imperative for the endocytic degradation and thus for TLR-stimulation. On this basis, an underlying hypothesis was that, depending on the bacterial stimuli, a reduction of the ROS by a stilbenoid hampers bacterial degradation and thus cytokine production.

We used LPS microbially induced ROS production as a simple and reliable strategy in our hands.

The final conclusion in the abstract states:

“In summary, the anti-inflammatory and the antioxidant activity showed no straight relationship, and appeared related to several factors, such as the type of the pro-inflammatory signal, and the chemical structure and bioavailability of the stilbenoids.”

None of the “several factors” including the pro-inflammatory signals or the chemical structure were clearly identified and bioavailability was not addressed. Thus, none of these conclusions were supported by the presented data. Since the two effects, the anti-inflammatory and anti-oxidative effects of stilbenoids, “showed no straight relationship” and the aforementioned conceptual shortcomings, the study is of poor scientific value

We agree that this part of the abstract does not adequately summarize our findings. We have thus changed it to:

“In summary, the immune modulating effect of the stilbenoids depends on both specific structural features of tested compounds and the stimulating bacteria. Interestingly, in the case of monomers we observed a straight correlation with their antioxidative activity, which was lost for dimers “.

In summary, I cannot support publication of this manuscript in IJMS and have thus no further specific comments

The revised version has been modified to accommodate the criticism of the reviewer. We believe that the manuscript now more clearly describes the novelty of our findings and possesses the sufficient quality to be published on IJMS.

Minor point:

Since BM-DC were generated from mice the authors should follow the “Ethical Guidelines for the Use of Animals in Research” (see Instructions for Authors) and provide details in the Material and Methods section.

We have included the following in the description:

“All animals used as source of bone marrow cells were housed under conditions approved by the Danish Animal Experiments Inspectorate (Forsøgdyrstilsynet) according to The Danish Animal Experimentation Act; LBK no. 474 from 15/05/2014, and experiments were carried out in accordance with the guidelines of ‘The Council of Europe Convention European Treaty Series (ETS)123 on the Protection of Vertebrate Animals used for Experimental and other Scientific Purposes’. ” (l. 319-325)

Round 2

Reviewer 2 Report

In the revised manuscript presented by Johnsen et al. the authors have only made editorial changes, which contribute to an improvement in comprehensibility, but do not change my fundamental criticism.

The authors admit:

We agree that the data are descriptive in nature and that we did not directly investigate the mechanisms behind the effect of the stilbenoids tested in this study.

And further:

Our data indicate that there is no straight relationship between the immune modulating effect of the stilbenoids and their antioxidant activity. All monomers except trimethoxy-resveratrol inhibited L. acidophilus NCFM induced IL-12, IL-10, and TNF-α production, while dehydro-δ-viniferin increased the IL-12 production. Despite having moderate to high total antioxidant activity, the two dimeric stilbenoids were weak inhibitors of LPS-induced ROS formation. However, as remarked in the discussion we see a correlation between the ROS inhibition by the monomeric stilbenes and the inhibition of IL-12 in the gram-positive stimulated bmDCs. Whether this reflects the inhibited ROS production or effects on signalling molecules is however not elucidated.

My major concern that the data presented is merely descriptive and does not provide any significant new mechanistic or conceptual insights has not changed substantially.

The authors responded to the criticism of the lack of a hypothesis:

Our main hypothesis for the work presented in this manuscript was that the effect of the various stilbenoids may depend on the bacterial stimuli and thus stilbenoids may affect the type of immune response induced by the bacteria (e.g. abrogation of the IL-12 may inhibit development of a Th1 cells needed for an efficient elimination of the bacteria). This has now been clearly stated in the abstract.

This would have been a hypothesis to test, showing that the efficient Th1 response (=mechanistic insight) and elimination of bacteria is impaired. The statement in the abstract is therefore only pure speculation that is not supported by the data.

The final conclusion in the abstract was now changed from In summary, the anti-inflammatory and the antioxidant activity showed no straight relationship,...” to “…in the case of monomers we observed a straight correlation with their antioxidative activity, which was lost for dimers“.

Without the data having changed, the opposite conclusion suddenly emerges? This proves impressively that the few data of this study cannot really support a meaningful and scientifically valubale conclusion.